# Nonlinear Thermopower Behaviour of N-Type Carbon Nanofibres and Their Melt Mixed Polypropylene Composites

**DOI:** 10.3390/polym14020269

**Published:** 2022-01-10

**Authors:** Antonio J. Paleo, Beate Krause, Maria F. Cerqueira, Enrique Muñoz, Petra Pötschke, Ana M. Rocha

**Affiliations:** 12C2T-Centre for Textile Science and Technology, Campus de Azurém, University of Minho, 4800-058 Guimaraes, Portugal; amrocha@det.uminho.pt; 2Leibniz-Institut für Polymerforschung Dresden e.V. (IPF), Hohe Str. 6, 01069 Dresden, Germany; krause-beate@ipfdd.de (B.K.); poe@ipfdd.de (P.P.); 3INL-International Iberian Nanotechnology Laboratory, Av. Mestre Jose Veiga, 4715-330 Braga, Portugal; fcerqueira@fisica.uminho.pt; 4CFUM-Center of Physics, Campus de Gualtar, University of Minho, 4710-057 Braga, Portugal; 5Facultad de Física, Pontificia Universidad Católica de Chile, Santiago 7820436, Chile; munoztavera@gmail.com

**Keywords:** polypropylene, carbon nanofibers, thermoelectric properties, n-type polymer composites, variable range hopping

## Abstract

The temperature dependent electrical conductivity *σ* (*T*) and thermopower (Seebeck coefficient) *S* (*T*) from 303.15 K (30 °C) to 373.15 K (100 °C) of an as-received commercial n-type vapour grown carbon nanofibre (CNF) powder and its melt-mixed polypropylene (PP) composite with 5 wt.% of CNFs have been analysed. At 30 °C, the *σ* and *S* of the CNF powder are ~136 S m^−1^ and −5.1 μV K^−1^, respectively, whereas its PP/CNF composite showed lower conductivities and less negative S-values of ~15 S m^−1^ and −3.4 μV K^−1^, respectively. The *σ* (*T*) of both samples presents a d*σ*/d*T* < 0 character described by the 3D variable range hopping (VRH) model. In contrast, their *S* (*T*) shows a d*S*/d*T* > 0 character, also observed in some doped multiwall carbon nanotube (MWCNT) mats with nonlinear thermopower behaviour, and explained here from the contribution of impurities in the CNF structure such as oxygen and sulphur, which cause sharply varying and localized states at approximately 0.09 eV above their Fermi energy level (*E_F_*).

## 1. Introduction

Thermoelectric (TE) materials are able to transform waste heat into electrical energy. Their efficiency is characterized by the dimensionless figure of merit $zT=\frac{S^{2}\sigma }{k}\;T$ where the thermopower (Seebeck coefficient) (*S*), calculated as *S* = Δ*V*/Δ*T*, reflects the creation of a potential difference (Δ*V*) when the ends of a TE material are exposed to a temperature difference, *T* is the absolute temperature, and *k* is the thermal conductivity [1,2]. Thereby, *S* can be positive or negative depending on the type of the majority charge carrier. Hence, in p-type TE materials (positive *S*), there is dominant hole conduction, whereas in n-type TE materials (negative *S*), the majority of the charge carriers are electrons [3]. In addition, the power factor PF = $S^{2}\sigma$ is used as a characteristic measure with *σ* as the electrical conductivity [2,4]. Accordingly, materials with large *S*, high *σ* and low *k* values are necessary for the achievement of high TE properties. In this respect, TE materials based on inorganic materials such as BiTe, SnSe and GeSe are very promising as they show *zT* values ≥ 1 [5]. However, the combination of their difficult processibility together with their high cost, partial toxicity, geopolitical risks and poor flexibility have increased interest in searching for different TE materials [6]. In this context, organic materials mainly including conducting polymers (CPs) such as poly(3,4-ethylene dioxythiophene) (PEDOT), polyaniline (PANI), polypyrrole (PPy) and their composites have observed fast progress [7]. Moreover, inorganic/organic hybrids, consisting of those electrically conductive polymers and inorganic TE materials, have also attracted great interest [8,9]. However, these type of compounds have not yet provided *zT*s higher than those obtained only with inorganic materials due to the random dispersion of the latter ones in the CPs, as well as the existing divergence of their Fermi energy levels (*E_F_*) [10]. Moreover, the processing in the melt state is more environmentally friendly than in solution as it permits the production of large material volumes with existing equipment and prevents the need of using solvents [11]. Consequently, conductive polymer composites (CPCs), consisting of insulating polymers (IPs) modified with carbon based conductive structures (i.e., carbon black, carbon nanotubes, graphene, etc.), are becoming promising due to the good balance yielded by the flexibility and low *k* provided by the IP and the high *σ* and *S* of the carbon materials [6,12]. Among the different types of carbon structures, most attention has focused on the investigation of the TE properties of single-walled carbon nanotubes (SWCNTs) and multi-walled carbon nanotubes (MWCNTs) due to their high *S* and the possibility of increasing their S-values by different processes [13,14]. Thus, melt-mixed composites of polybutylene terephthalate (PBT) with 5 wt.% SWCNTs have achieved positive S values of 66 μV K^−1^, whereas negative S-values of −51.5 µV/K have been reported in acrylonitrile butadiene styrene (ABS) composites with 0.5 wt.% SWCNTs [14]. On the other hand, nitrogen doped MWCNTs always resulted in negative S-values of their composites [15]. Despite the enormous interest in CNTs as novel TE materials mentioned above, very limited efforts have focused recently on the TE properties of a different carbon nanostructure known as carbon nanofibres (CNFs), which typically present larger diameters than CNTs and diverse orientation of the graphitized shells with respect to their hollow tubular axis [16]. This is surprising since early works on heat-treated benzene-derived carbon fibres prepared by thermal decomposition [17], heat-treated graphite fibres grown by pyrolysis of natural gas [18], and heat-treated methane-derived vapor grown carbon fibres (VGCFs) [19], have shown air-stable negative S values at room temperature, contrary to the most as-produced CNTs, which are p-type conductors due to oxygen doping [20]. More recently, a thermopower of −5.5 µV/K at 30 °C has been reported for a selected commercial grade of CNFs [21] that additionally showed the benefit of also having air-stable negative S-values. It must be noted that such n-type materials are also needed to fabricate effective thermoelectric generators (TEGs) which typically consist of pairs of p- and n-type materials. It is in this context that this study was performed, extending the TE analysis of former works focused on n-type CNFs and their melt-mixed CNF/polypropylene (PP) composites [21,22]. Thereby, in those previous works, the TE properties of n-type CNFs (Pyrograf^®^-III PR 24 LHT XT) and their melt-mixed PP composites with up to 5 wt.% of CNFs are analysed and compared exclusively at room temperature. In the present work, the *σ* (*T*) and *S* (*T*) values in the temperature interval from 30 °C to 100 °C of a different grade of n-type CNFs (Pyrograf^®^-III PR 19 LHT XT), and their melt-mixed PP/CNF composite with only 5 wt.% of CNFs are studied, compared and theoretically modelled. To our knowledge, the *S* (*T*) of such CNFs and their melt-mixed polymer composites has not been presented before.

## 2. Materials and Methods

### 2.1. Material Processing

A polypropylene powder, Daplen™ EE002AE (Borealis AG, Vienna, Austria)—a reactor elastomer modified polypropylene intended for injection moulding with a density of 905 kg m^−3^ and melt flow rate of 11 g 10 min^−1^ (230 °C/2.16 kg)—was used as polymer matrix. Pyrograf^®^-III PR 19 LHT XT carbon nanofibres (ASI, Cedarville, OH, EUA) with bulk densities between 1–3 lb/ft^3^ (0.016–0.048 g cm^−3^) and a range of lengths of 30–100 µm produced by chemical vapour deposition (CVD) were chosen for producing the melt-mixed PP/CNF composites. This particular type of CNF was grown at 1100 °C with a thermal post-treatment at 1500 °C in inert atmosphere, which morphologically results in a dual wall structure surrounding the hollow tubular core as it is shown in Figure 1.

Melt-mixed PP/CNF composites were fabricated on a modular lab-scale intermeshing mini-corotating twin-screw extruder, with a screw diameter of 13 mm, barrel length of 338 mm and an approximate L/D ratio of 26 coupled to a cylindrical rod dye of approximately 2.85 mm diameter. Further description of the processing conditions has been previously reported [23]. The extruded PP/CNF composites were pelletized and compression-moulded at 210 °C with a PW40HT hot press for 2 min (1.5 min pre-heating, maximal force 50 kN, 0.5 min cooling in a mini-chiller, with a polyimide foil as separator). Circular films with a diameter of 60 mm and a thickness of 0.5 mm were prepared for thermoelectric measurements and circular pieces with 12.5 mm diameter and 1 mm thickness for thermal conductivity measurements. Based on the electrical conductivity values, in addition to the Pyrograf^®^-III PR 19 LHT XT CNF powder, the melt-mixed PP/CNF composite with 5 wt.% CNFs (above the electrical percolation threshold [23]) was selected for the morphological, structural and thermopower analysis.

### 2.2. TEM, SEM, Raman and XPS Analysis

The as-received CNFs were observed with a transmission electron microscope (TEM) (JEM-2100, JEOL Ltd., Tokyo, Japan) operating a LaB6 electron gun at 80 kV. Images were acquired with an “OneView” 4k × 4k CCD camera at minimal under-focus to achieve visibility of the CNF surface layers. Extruded strands of the PP/CNF composite were cryo-fractured in liquid nitrogen and the surface was covered with 3 nm platinum before examining using field emission scanning electron microscopy (SEM) (Ultra plus, Zeiss Oberkochen, Germany) at 3 kV. Raman spectroscopy measurements (ALPHA300 R Confocal Raman Microscope, WITec GmbH, Ulm, Germany) were carried out using 532 nm laser for excitation in back scattering geometry. The laser beam with P = 0.5 mW was focused on the sample by a 50× lens (Zeiss), and the spectra were collected with 600 groove/mm grating using 5 acquisitions with 2s acquisition time. The surface characterization was performed by means of X-ray photoelectron spectroscopy (XPS) (ESCLAB 250Xi, Thermo Fisher Scientific, Waltham, Massachusetts, EUA) in the ultra-high vacuum (UHV) system. The base pressure in the system was below 5 × 10^−10^ mbar. The XPS spectra were acquired with a hemispherical analyser and X-ray source producing monochromated Al Kα (hν = 1486.61 eV) radiation operated at 15 kV, power 200 W and X-ray beam spot size 0.65 mm. They were recorded with pass energies of 20 eV and 200 eV for high resolution and survey spectra, respectively. The XPS spectra were peak-fitted using the Avantage processing software (Thermo Fisher Scientific). The Lorentzian/Gaussian (30/70%) line shape and “Smart” background subtraction were used for peak fitting, and the quantification has been done using sensitivity factors provided by the Avantage library.

### 2.3. Thermoelectric Analysis

The Seebeck coefficient and volume resistivity of the PP/CNF composite and CNF powder were determined using the self-constructed equipment TEG at Leibniz-IPF [24,25]. The measurements were performed at the mean temperatures of 303.15 K (30 °C), 313.15 K, 333.15 K, 353.15 K and 373.15 K (100 °C), getting the S values by applying temperature differences between the two copper electrodes up to ±8 K around the mean temperature in 2 K steps. The samples were painted with a conductive silver ink at their ends. The measurements of thermovoltage and resistance were performed using a Keithley multimeter (DMM2011, Keithley Instruments, Cleveland, OH, EUA). The volume resistivity was measured at the different mean temperatures using a 4-wire technique. The given values represent the arithmetic mean values of ten measurements. The Seebeck coefficient at each temperature was calculated as the average of eight thermoelectric voltage measurements. For the thermoelectric measurements on the CNF powder, an insert consisting of a PVDF tube (inner diameter 3.8 mm, length 16 mm) closed with copper plugs was used, which was filled with the CNF powder [14].

The thermal conductivity of the composites was calculated from the product of thermal diffusivity, density, and specific heat capacity. The thermal diffusivity was measured on circular samples (diameter 12.5 mm, thickness 1 mm) through the plate thickness using the light flash apparatus (LFA 447 NanoFlash, Netzsch-Gerätebau GmbH, Selb, Germany) at 303.15 K, 313.15 K, 333.15 K, 353.15 K and 373.15 K. The specific heat capacity of the composites was calculated by comparing the signal heights between the composite and the reference Pyroceram 9606 (with known specific heat capacity) using the LFA 447 NanoFlash software. The density of the composites was determined using the buoyancy method. The given values represent the mean values of four measurements.

## 3. Results and Discussion

### 3.1. TEM, SEM, Raman and XPS Analysis

Representative TEM images of the used CNFs are shown in Figure 1. The diameter of 25 CNFs was measured and an average diameter of around 110 nm was obtained. This diameter is comparable to the values measured by Tessonnier et al. for the same type of CNFs [26]. The CNFs show a two-layer structure surrounding the hollow tubular core, where the inner layers present parallel nanosheets with respect to the hollow core. In particular, the inner layers of Figure 1b were composed of 29 nanosheets with individual thicknesses of about 0.32 nm and angles of around 17° with respect to the main axis of the CNF. Similar nanosheets are also present in the outer layers, though they are not as ordered as in the case of the internal layers. In total, 19 outer graphene sheets with dimensions between 0.32 and 0.35 nm are observed in Figure 1b. As a conclusion, Figure 1a,b showed two types of morphologies enclosing the hollow tubular core. Moreover, the inner layers seem to be better graphitized than the outer ones.

SEM micrographs related to the PP/CNF composite are shown in Figure 2. The CNFs protrude from the PP and are seen in relatively large length, which is a sign of a low wettability and poor adhesion. This may be due to the not favourable affinity between the nonpolar polyolefin matrix and the CNFs. However, the CNFs seen in Figure 2b indicate a homogenous dispersion and distribution without the presence of agglomerates.

Figure 3 shows the Raman spectra obtained from PP, as-received CNF powder and the PP/CNF composite in the range between 600 and 1800 cm^−1^. Polypropylene shows rich Raman spectra with modes in the range 800–1500 cm^−1^ [27] assigned to CH_n_ stretching vibrations [28]. On the other hand, the as-received CNFs present the two bands expected in carbon nanostructures. One band is found at 1352 cm^−1^, known as the disordered-induced D band [29], observable when defects are present in the carbon aromatic structure, and the G-band is found at 1580 cm^−1^, characteristic of the ideal graphitic lattice vibration [30]. As expected, the PP/CNF composite presents the signatures of the two base materials. In particular, the most intense modes of PP (dotted lines in Figure 3a) are clearly observable in PP/CNF composites. Interestingly, a relative intensity change is observed between the two PP peaks located at the vicinity of 800 cm^−1^, which can be related to the presence of the CNFs in the PP/CNF composite. Moreover, the peak at 1460 cm^−1^, corresponding to the PP (numbered as 3 in Figure 3a) also shows the same intensity as the G and D bands in the PP/CNF composite, whereas peaks 1 and 2 of the PP hide the D-band of the CNFs in the PP/CNF composite (Figure 3b). These results suggest a strong presence of PP in the analysed area. It is also of note that the G peak for the PP/CNF composite shows an apparent shoulder peak at ~1600 cm^−1^, which can be associated to the existence of a D’ mode caused by defects in the CNFs [31]. The peak position and the full width half maximum (FWHM) of the modes for CNFs and the PP/CNF composite were determined by fitting the Raman spectra with Lorentzian functions, and the obtained parameters are listed in Table 1 together with the in-plane graphitic domain size (La) calculated according to L_a_ (nm) = 4.4/(I_D_/I_G_) [32]. Interestingly, the FWHM_G_ (90) and FWHM_D_ (115) of the CNF powder are higher than previous values reported for the same type of CNFs [26]. Typically, a decrease in the width of the D and G bands has been associated with an enhancement of the degree of graphitization in CNFs [26]. Notably, the values of FWHM_G_ and FWHM_D_ decrease clearly in the PP/CNF composites, an observation that has been attributed earlier to the increasing size of remaining agglomerates when enhancing the concentration of CNFs [31]. However, such effect was not observed based on the SEM images shown in Figure 2. Table 1 shows that the G and D peak positions were practically the same and with slightly higher wavenumbers than the CNF powder in the PP/CNF composite. This indicates that the processing did not affect seriously the structure of the as-received CNFs. The intensity ratios between the D and G bands (I_D_/I_G_) were calculated and presented also in Table 1 since they are an important parameter for quantifying the number of disordered (D) and ordered (G) carbon atoms [33]. Notably, the I_D_/I_G_ of 0.76 for the CNF powder results lower than the value of 1 reported for the same type of CNFs [26]. Moreover, the I_D_/I_G_ remains practically the same for the CNF powders and the PP/CNF composites, and, therefore, the production of the PP/CNF composites by melt-mixing did not have a strong effect on this parameter. However, the calculated in-plane graphitic domain size L_a_ was slightly higher for the PP/CNF composite. This result agrees with a previous work on melt-mixed PP/CNF composites made with another grade of Pyrograf^®^-III CNFs (PR 24 LHT XT), and it can be explained by the increase of the PP crystallite size caused by the introduction of CNFs [22].

The elemental composition of PP, as-received CNFs, and melt-mixed PP/CNF composite was analysed by XPS, and listed in Table 2. All samples contain mainly carbon and oxygen, as is evidenced by their survey XPS spectra (Figure 4). In addition, some inorganic impurities such as sulphur (S) (~0.1%) in the as-received CNF powder, and silicon (Si) (~0.5%) in PP and the PP/CNF composite (Table 2) were detected. In particular, the presence of sulphur in this type of CNFs has been previously observed, together with growth catalyst impurities such as iron [26]. The presence of Si in PP and the PP/CNF composite can be assigned to contaminations accumulated during the melt-mixing process. Moreover, the composition analysis of the neat CNFs revealed an amount of oxygen of ~1.62%, which increases up to ~2.64% in the PP/CNF composite (Table 2).

A comparison of the deconvolution of C1s and O1s spectra for PP, neat CNFs and the melt-mixed PP/CNF composite is presented in Figure 5. The C1s deconvolution of the neat PP shows a strong line at ~285.8 eV assigned to C–C bonds, and another line at 286.5 eV assigned to C–O bonds (Figure 5e) [34]. In the case of the neat CNFs, their C1s show a strong line at ~284.4 eV assigned to C–C bonds, which together with the “satellite” peaks at 290.5 eV represent sp^2^ hybridized carbon, an additional peak is observed at 285.2 eV assigned to C–O bonds (Figure 5c) [35,36]. The C1s deconvolution of PP/CNF composites presents similar results as the neat CNFs, though the C–C and C–O bonds are slightly shifted to 284.8 eV and 285.3 eV, respectively. However, it is of note that the π–π^*^ peaks did not appear in the C1s spectra of PP/CNF composites (Figure 5a), which can be explained by the high presence of PP. On the other hand, the O1s deconvolution of neat PP shows a strong line at 532.7 eV (Figure 5f) assigned to C–O bonds [37]. The O1s spectra in CNFs (Figure 5d) yielded asymmetric peaks at 531.6 eV assigned to C=O [38], and 533.2 eV assigned to C–O [39]. Interestingly, the O1s spectra of PP/CNF composites (Figure 5b) show only a peak at 532.2 eV, which can be associated to both C–O and C=O bonds.

### 3.2. Thermoelectric Analysis

The electrical conductivity of the CNF powder and the PP/CNF composite as functions of the measuring temperature are presented as squared symbols in Figure 6, whereas the exact values can be seen in Table 3. In particular, a value of 136.39 ± 0.22 S m^−1^, equivalent to ~7 × 10^−1^ Ohm cm, is measured for the CNF powder at 303.15 K (30 °C), which decreases up to 127.01 ± 14.36 S m^−1^ at 373.15 K (100 °C). The electrical resistivity at 30 °C is significantly higher than the value of 4 × 10^−3^ Ohm cm previously reported for individual Pyrograf^®^ III CNFs [40]. This difference can be ascribed to the higher electrical resistivity, originated by higher contact resistances at the connection points of the nanofibres within the compressed CNF powder. Interestingly, the electrical conductivity of the here-used CNF grade is lower than the values of ~320 S m^−1^ at 30 °C reported for the Pyrograf^®^ III CNF grade PR 24 LHT XT as previously reported [21]. This significant difference between the *σ* of both CNF grades could be explained by the higher agglomerate sizes reported of the as-received PR 19 LHT XT CNFs, ranging from 100 to 200 µm as compared to 10 to 100 µm observed for the as-received PR 24 LHT XT CNFs [26]. In this respect, the higher ordered outer layers of the latter CNF grade (PR 24 LHT XT) together with the lower diameters of its outer layers could enhance not only the electron transport within and across their sidewalls, but also the electronic hopping between adjacent CNFs, thus explaining their higher *σ* values. Notably, d*σ*/d*T* < 0 is observed over the studied range of temperatures, which is in contrast to previous works reporting the temperature dependence electrical conductivity of SWCNT [41,42,43] and MWCNT mats [44]. The *σ* (*T*) of the CNF powder has been described in this study by the 3D variable range hopping (VRH) model [41,45]:(1)$$\sigma \left(T\right)=\sigma_{0\;}\;\exp[-(\frac{T_{C}}{T}){\;}^{\frac{1}{4}}\;]$$
where *σ*_0_ is considered as the value of conductivity at an infinite temperature, while *T_C_* is the characteristic temperature that determines the thermally activated hopping among localized states at different energies. In particular, *T_C_* can be expressed as function of the energy barrier constant (*W_D_*) as $T_{C}\equiv \;\frac{\left|W_{D}\right|}{k}$, with *k* as the Boltzmann´s constant. A good agreement is observed for the CNF powder by fitting the Equation (1) with$\;\sigma_{0}=46.40\;{S\;m}^{-1}$, $T_{C}=393.2\;K$, and $W_{D}=-34\;\mathrm{meV}$ (Figure 6). Interestingly, the value of *T_C_* is in the same order as some reported SWCNT mats (250 K) [43], though more recent works have reported *T_C_* values with one order of magnitude lower (20 K) for SWCNT mats tested in the 0–300 K interval [41,42]. Notably, the negative sign of *W_D_* can be explained by the presence of impurities such as the iron (Fe) (1.19%) detected in another study by wavelength dispersive X-ray fluorescence (WDXRF) [26] and the oxygen (1.6%) and sulphur (0.1%) contents observed by XPS in this study. These impurities could originate a thermal-enhanced backscattering mechanism due to the presence of virtual bound-states, represented as sharp peaks near the E_F_ in the density of states [44,46]. This is an unexpected phenomenon since the majority of work that evaluates the *σ* (*T*) of CNT mats reports positive signs of *W_D_* [41,44]. In comparative terms, the barrier energy value of the CNF powder (34 meV) is in the middle between the activation energy reported for n-type graphitized carbon fibres (60 meV) in the 250–750 K interval [47], and the activation energy reported for SWCNT mats doped with boron (12.2 meV) in the 0–320 K range by using the Arrhenius approach [44].

Similar to the CNF powder, the PP/CNF composite shows a decrease in their conductivity (d*σ*/d*T* < 0) from 15.36 ± 0.01 S m^−1^ at 303.15 K (30 °C) to 13.86 ± 0.45 S m^−1^ at 373.15 K (100 °C) (Table 3). It must be noted that the PP/CNF composite is above its electrical percolation threshold (*ϕ_c_*). For the PP/4 wt.% CNF composite, a volume resistivity of around 10^7^ Ohm cm was measured which was too high for performing stable TE measurements. In an earlier study the *ϕ_c_* was found to be between 1 and 2 wt.%, when using the same mixtures, but compression moulded under slightly different conditions [48]. The σ of 15.36 S m^−1^ for the PP/CNF composite is significantly lower than the σ of the CNF powder (136.39 S m^−1^) at 303.15 K. This significant difference, typically observed in melt-mixed polymer composites, can be attributed to the wrapping of polypropylene chains around the CNFs, which increases the contact resistance between the adjacent CNFs, resulting in the rise of the CNF network resistivity [49]. Notably, the σ of PP/CNF composites (15.36 S m^−1^) was lower than the σ of melt-mixed PP composites with 5 wt.% of the PR 24 LHT XT CNFs grade (49.5 S m^−1^) reported previously [21], which can be related to the structural differences and higher conductivities observed for the PR 24 LHT XT CNF powder. Moreover, the *σ* (*T*) for the PP/CNF composites has been also depicted by the 3D VRH model (Equation (1)), from which$\;\sigma_{0}=1.76\;{S\;m}^{-1}$, $T_{C}=7.43\times 10^{3}\;K$ and $W_{D}=-640\;\mathrm{meV}$ have been obtained. Thereby, $\sigma_{0}\;$is one order of magnitude lower than the $\sigma_{0}\;$of the CNF powder due to the insulating nature of the matrix PP. Interestingly, a similar *σ*_0_ of 1.75 S m^−1^ by using the Arrhenius equation has been reported for epoxy composites filled with 10 phr of MWCNTs tested in the 303.15–433.15 K range [50]. However, the obtained *T_C_* resulted two orders of magnitude lower than the *T_C_* reported for epoxy composites filled with 35 phr of carbon black (CB) (6.19 × 10^5^ K) tested in the range 303.15–453.15 K by using the 3D VRH [51]. This implies that, the thermally activated backscattering mechanism can again be responsible for the negative *W_D_* found in the PP/CNF composite. Notably, the d*σ*/d*T* < 0 contrasts with the d*σ*/d*T* >0 (in the 0–300 K interval) reported in epoxy composites produced with exactly the same PR 19 LHT XT CNFs [52]. Furthermore, the absolute value of *W_D_* (0.640 eV) is consistent with the activation energy (0.488 eV) obtained by using the Arrhenius approach for epoxy composites filled with 5 phr MWCNTs [50]. In view of these results, it is remarked that a statistical model such as VRH seems to provide a good first approach to understand the underlying transport mechanism behind the high degree of morphological disorder present in the materials studied in this work (as-received CNF powder and PP/CNF composite).

The triangle symbols in Figure 6 present the thermopower of the CNF powder and the PP/CNF composite as a function of temperature. The n-type character of the CNF powders is found at all temperatures, and it contrasts to the findings of most as-produced CNTs that are p-type due to their oxygen doping with the environment [53]. In particular, the CNF powder presented an S-value of −5.10 ± 0.03 μV K^−1^ at 303.15 K (30 °C), which increases gradually (in absolute value) up to −5.83 ± 0.03 μV K^−1^ at 373.15 K (100 °C) (Table 3). Notably, that S-value (−5.1 μV K^−1^) is slightly lower than the value of −5.5 ± 0.1 μV K^−1^ at 303.15 K reported for the Pyrograf^®^ III CNF PR 24 LHT XT grade [21]. Moreover, the results are similar to the scarce negative S-values found in literature for free-standing MWCNT films grown by CVD at room temperature [54], and the −6 μV K^−1^ reported for MWCNT buckypapers grown by CVD [55]. The *S* (*T*) of the CNF powder can be depicted by the theoretical model proposed by Choi et al. for describing the nonlinear thermopower behaviour of doped MWCNT mats [44]:(2)$$S\;\left(T\right)=bT+\frac{cT_{p}}{T^{2}}\frac{\exp\left(\frac{T_{P}}{T}\right)}{{\left[\exp\left(\frac{T_{P}}{T}\right)+1\right]}^{2}}$$

In Equation (2), *bT* represents the metallic (linear) term, *c* is a constant, and $T_{p}=\left(E_{p}-E_{F}\right)/k$ where *k* is the Boltzmann´s constant, *E_F_* is the Fermi energy level, and *E_P_* is the energy corresponding to the sharply varying and localized states near *E_F_* in the density of states due to the contribution of impurities [44,46]. The best fit of the *S* (*T*) with Equation (2) for the CNF powder (Figure 6) shows that the first term is positive with $b=5.50\times 10^{-3}\;{\mu V\;K}^{-2}$, while the second term is negative with $c=-1.80\times 10^{4}\;\mu V\;K$ and $T_{p}=993.21\;K$, yielding a $E_{P\;}-E_{F}=0.086\;\mathrm{eV}$ and an overall negative S that gradually increases (in absolute value) with temperature. Thus, for instance at 303.15 K, the metallic contribution yields ~+1.7 μV K^−1^, and the second term is ~−6.8 μV K^−1^, whereas at 373.15 K, the metallic contribution is ~+2 μV K^−1^ and the second term is ~−7.8 μV K^−1^.

The PP/CNF composite shows thermopowers from −3.44 ± 0.03 μV K^−1^ at 303.15 K to −4.34 ± 0.14 μV K^−1^ at 373.15 K. The *S* (*T*) of the PP/CNF composite is also fitted by Equation (2) with the best fit resulting in $b=1.55\times 10^{-3}\;{\mu V\;K}^{-2}$, $c=-1.29\times 10^{4}\;\mu V\;K$, $T_{p}=1083.9\;K$ and $E_{P\;}-E_{F}=0.094\;\mathrm{eV}$. Thus at 303.15 K, the metallic contribution is ~+0.4 μV K^−1^, and the second term is ~−4.0 μV K^−1^, whereas at 373.15 K, the metallic contribution is ~+0.6 μV K^−1^ and the second term is ~−5.0 μV K^−1^. Therefore, as in the case of the CNF powder, the overall sign of the thermopower in the PP/CNF composite is dominated by the resonances near the *E_F_* at the density of states (second term of Equation (2)) caused by impurities present in the CNF structure. It is observed by comparing the results at 303.15 K that the positive metallic contribution of the PP/CNF composite (+0.4 μV K^−1^) is ~77% lower than that of the CNF powder (+1.7 μV K^−1^), and this difference is slightly reduced at 373.15 K (~70%). This result can be explained as a consequence of the oxygen doping reduction in the CNFs caused by the presence of the insulating PP chains. Similarly, it is also of note that at 303.15 K the second term of Equation (2) is ~41% lower in absolute value for the PP/CNF composite (−4.0 μV K^−1^) than that of the CNF powder (−6.8 μV K^−1^). This change is again reduced at 373.15 K (~36%). Thus, the presence of PP chains around the nanofibres seems to clearly affect the second term of Equation (2). In this respect, the slight electron transfer from the CNF outer layers towards the surrounding PP molecular chains [21] could explain the lower absolute values of the second term in Equation (2) found for the PP/CNF composite.

The power factor PF as function of temperature of the PP/CNF composite and CNF powder were calculated, and the results are shown in Table 3. At 303.15 K, the CNF powder shows a PF of 3.5 × 10^−3^ μW·m^−1^·K^−2^, whereas the PP/CNF composite achieves a PF of 1.8 × 10^−4^ μW·m^−1^·K^−2^ (lower than the PF of 7.0 × 10^−4^ obtained in PP/CNF composite with 5 wt.% of PR 24 LHT XT CNFs [21]). The PF increases in both materials at 373.15 K as a consequence of its proportionality to *S*^2^. The CNF powder achieves a value of 4.3 × 10^−3^ μW·m^−1^·K^−2^, whereas the PP/CNF composites achieves a PF of 2.6 × 10^−4^ μW·m^−1^·K^−2^. For comparison, PF values of ~10^−1^ μW·m^−1^·K^−2^ have been recently reported in a melt-mixed PP composite with 2 wt.% of p-type boron-doped SWCNTs [56].

The figure of merit *zT* of the PP/CNF composite increases slightly from 2.2 × 10^−7^ (30 °C) up to 4.4 × 10^−7^ (100 °C) due to the increasing PF and slightly decreasing thermal conductivity values (see Table 3). This zT value is lower compared to 4.8 × 10^−7^ (30 °C) obtained for PP/5 wt.% PR 24 LHT XT CNFs [21]. It must be also noted that higher zT values (3.0 to 3.3 × 10^−5^ at 40 °C) were reported for PP composites filled with 2 wt.% SWCNT Tuball^TM^ or 2 wt.% branched MWCNT CNS-PEG [14].

## 4. Conclusions

The electrical conductivity (*σ*) and thermopower (*S*) of as-received vapour grown carbon nanofibre (CNF) powder and its melt-mixed polypropylene composite with 5 wt.% of CNFs in the temperature range between 30 °C and 100 °C have been analysed in this study. At 30 °C, the *σ*, *S* and power factor (PF) of the as-received CNFs are ~136 S m^−1^, −5.1 μV K^−1^ and 3.5 × 10^−3^ μW m^−1^ K^−2^, respectively. The PP/CNF composite, for its part, shows lower conductivities of ~15 S m^−1^, and less negative S-values of −3.4 μV K^−1^, which correspond to a PF of 1.8 × 10^−4^ μW m^−1^ K^−2^ at 30 °C. Notably, the *σ* (*T*) of both materials presents a d*σ*/d*T* < 0 character, which is in contrast to the d*σ*/d*T* > 0 generally found for CNT mats and can be perfectly fitted by the statistical 3D VRH model. As regards their *S* (T), it follows the theoretical model proposed for describing the nonlinear thermopower of certain doped MWCNT mats. In particular, the d*S*/d*T* > 0 behaviour observed in both samples has been physically interpreted by the presence of some impurities in the CNFs, which could produce sharp peaks close to the Fermi energy level (*E_F_*) in their density of states. In summary, this study gives new insights in the origin of the negative and air stable S-values found in some commercial grades of as-received vapour grown carbon nanofibres and their melt-mixed composites with insulating polymers.

## Figures and Tables

**Figure 1 polymers-14-00269-f001:**
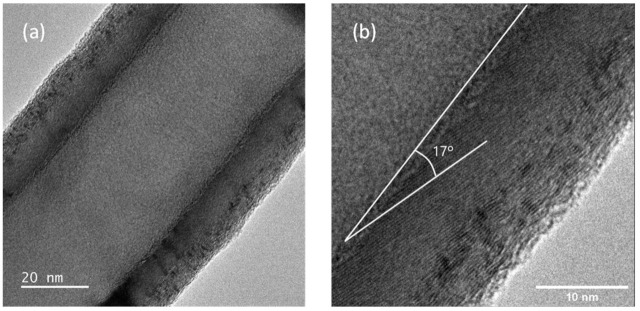
TEM images of vapor grown carbon nanofibres Pyrograf^®^-III PR 19 LHT XT, (**a**) hollow core and surrounding layers, (**b**) detail of inner layers constituted of parallel graphene sheets.

**Figure 2 polymers-14-00269-f002:**
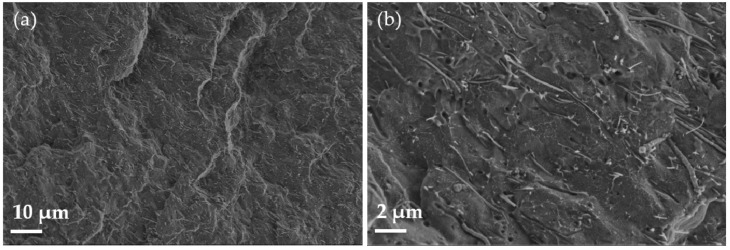
SEM micrographs of the PP/5wt.% CNF composite at lower (**a**) and higher (**b**) magnifications.

**Figure 3 polymers-14-00269-f003:**
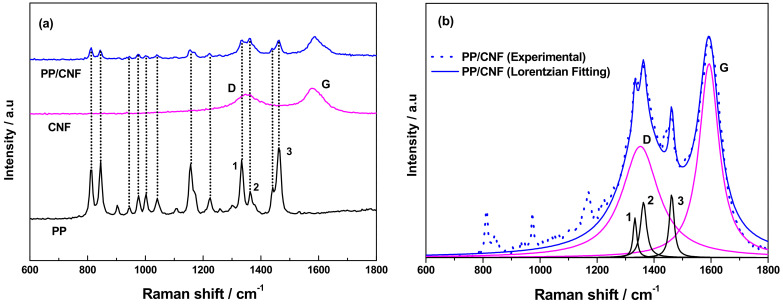
(**a**) Raman spectra of PP, as-received CNFs and PP/5wt.% CNF composite. (The dotted lines are to guide the eye), (**b**) Raman fit in the G and D wavenumber range of PP/5wt.% CNF composite (blue line), showing the different components: CNFs (violet line) and PP (black line).

**Figure 4 polymers-14-00269-f004:**
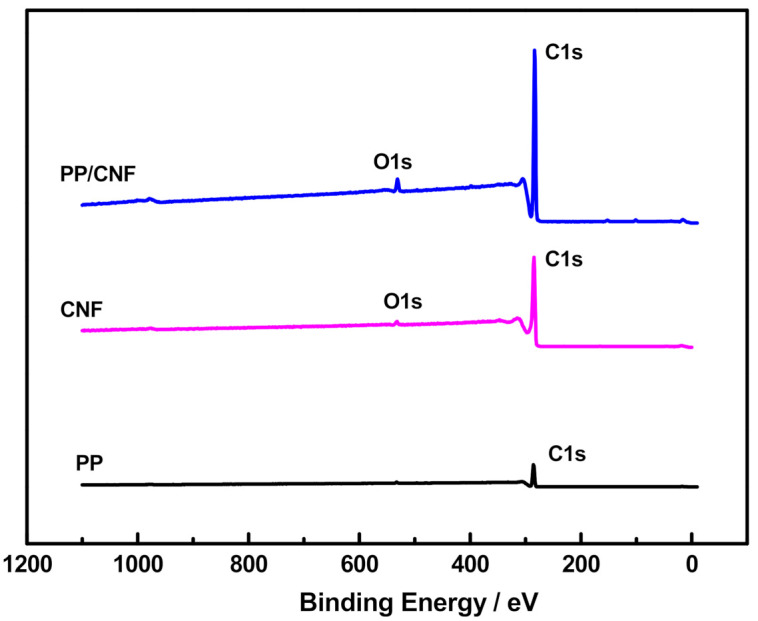
XPS survey spectra of PP, CNF powder, and PP/5 wt.% CNF composite.

**Figure 5 polymers-14-00269-f005:**
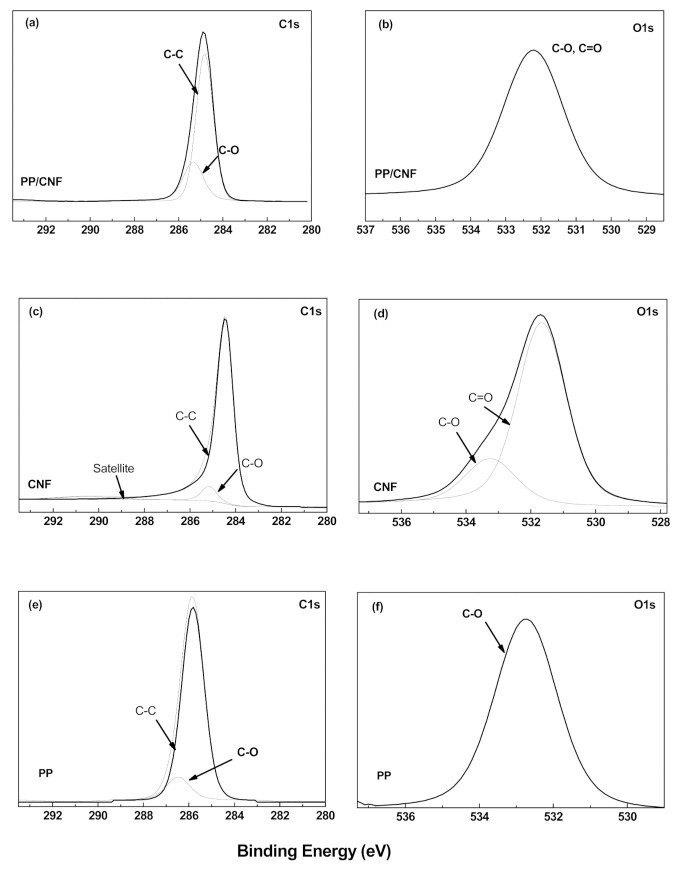
XPS deconvolution of C1s (**a**) and O1s (**b**) of PP/5 wt.% CNF composite; C1s (**c**) and O1s (**d**) of CNFs; C1s (**e**), O1s (**f**) of PP.

**Figure 6 polymers-14-00269-f006:**
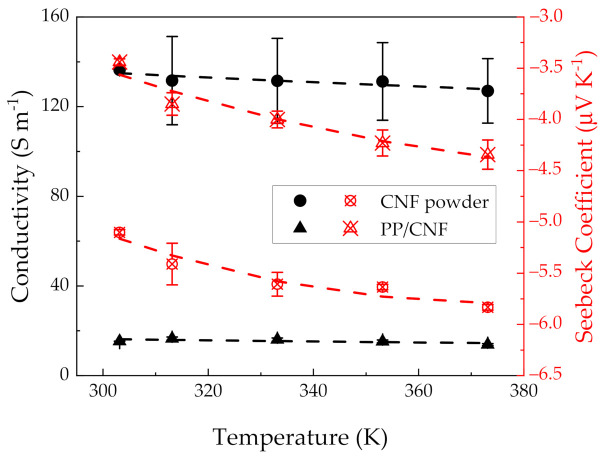
Electrical conductivity (squared symbols, black) and Seebeck coefficient (triangle symbols, red) as function of temperature of CNF powder and PP/5 wt.% CNF composite. The black and red dash lines represent the fitting with Equation (1) and Equation (2), respectively.

**Table 1 polymers-14-00269-t001:** D and G, FWHM, I_D_/I_G_, and La calculated according to [32], of as-received CNFs and PP/5wt.% CNF composite.

Sample	ω_G_ (cm^−1^)	FWHM_G_ (cm^−1^)	ω_D_ (cm^−1^)	FWHM_D_ (cm^−1^)	I_D_/I_G_	L_a_(nm)
CNF	1580	90	1352	115	0.76	5.8
PP/CNF	1587	50	1353	75	0.7	6.3

**Table 2 polymers-14-00269-t002:** Chemical composition and surface atomic % of PP, CNF and PP/5 wt.% CNF composite.

Sample	O1s	C1s	S2p	Si2p
PP	2.0	97.5	-	0.5
CNF	1.6	98.2	0.1	-
PP/CNF	2.6	96.9	-	0.5

**Table 3 polymers-14-00269-t003:** Thermoelectric properties of CNF powder and PP/5 wt.% CNF composite as function of temperature.

	CNF Powder			PP/CNF				
Temperature (K)	*σ* (S m^−1^)	*S* (µV K^−1^)	PF (μW m^−1^K^−2^)	*σ* (S m^−1^)	*S* (µV K^−1^)	PF (μW m^−1^K^−2^)	*k* (W m^−1^K^−1^)	*zT*
303.15	136.39 ± 0.22	−5.10 ± 0.03	3.55 × 10^−3^	15.36 ± 0.01	−3.44 ± 0.03	1.82 × 10^−4^	0.25	2.20 × 10^−7^
313.15	131.51 ± 19.70	−5.41 ± 0.20	3.85 × 10^−3^	16.54 ± 0.70	−3.85 ± 0.11	2.45 × 10^−4^	0.25	3.07 × 10^−7^
333.15	131.48 ± 18.95	−5.61 ± 0.12	4.13 × 10^−3^	16.07 ± 0.72	−4.00 ± 0.08	2.57 × 10^−4^	0.24	3.57 × 10^−7^
353.15	131.21 ± 17.33	−5.64 ± 0.03	4.17 × 10^−3^	15.26 ± 0.64	−4.23 ± 0.13	2.73 × 10^−4^	0.24	4.09 × 10^−7^
373.15	127.01 ± 14.36	−5.83 ± 0.03	4.32 × 10^−3^	13.86 ± 0.45	−4.34 ± 0.14	2.61 × 10^−4^	0.22	4.39 × 10^−7^

## Data Availability

The authors confirm that the data supporting the findings of this study are available within the article.

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
