# Peer review of "Nonlinear Thermopower Behaviour of N-Type Carbon Nanofibres and Their Melt Mixed Polypropylene Composites"

_polymers, 2022, doi:10.3390/polym14020269_

Round 1

Reviewer 1 Report

Paleo et al present a manuscript entitled "Nonlinear Thermopower Behaviour of N-Type Carbon Nano-fibers and Their Melt Mixed Polypropylene Composites".

The manuscript presents an extension of their own former works focused on n-type CNFs and their melt-mixed Carbon Nano-fibers (CNF)/polypropylene (PP) composites performed at room temperature [12,14]. In this work, the conductivity and Seebeck coefficient dependence with temperature in the 303-373K range is studied for their melt-mixed CNF/PP composites. In addition, structural and chemical characterization of the samples by means of Raman and X-ray photoelectron spectroscopies is provided.

In my opinion, the manuscript could be eventually accepted for publication in "Polymers" after revising some queries that are detailed in the following lines:

Q1) The Raman spectra discussion is quite complete but some brief discussion must be given regarding the possible errors on the calculated parameters such as "Id/Ia" and "La" parameters obtained particularly for the CNF/PP sample.

Q2) Additionally, an inset in Fig. 3 showing the deconvolution of the lorentzian peaks for the G and D peaks would be very clarifying.

Q3) Note that the PP Raman peaks 1 and 2 seem to modify their relative intensity for the CNF/PP sample. Although this could be affected by the overlapping with the D peak, a possible explanation on the relative intensity modification of PP peaks 1 and 2 could enrich the discussion.

Q4) Note also that the G peak for the CNF/PP sample shows an apparent shoulder peak at ~1600 cm-1. Is this possibly related to the increment of oxygen-containing groups observed in the XPS analysis?

Q5) In my opinion, the use of the VRH model to describe the electrical conductivity of these samples is quite questionable, particularly due to the obtained negative activation energies, which are not envisaged in references [18] nor [36,38]. Perhaps the problem is that the collection of 5 conductivity values in such a narrow temperature range is not enough to have strong evidence of the electrical conduction mechanism. Anyhow, isn't a metallic or a mixed semiconducting-metallic behavior possibly contributing in these samples?

Authors can refer to the following articles to enrich this part of the discussion:

-https://doi.org/10.1038/s41598-018-28043-3

-https://doi.org/10.1039/C5TC02053K

Reviewer 2 Report

The manuscript under the title: “Nonlinear thermopower behaviour of n-type carbon nanofibers and their melt mixed polypropylene composites” is in line with Polymers journal. This topic is relevant and will be of interest to the readers of the journal. It based on original research. This research has scientific novelty and practical significance. It based on original research. The article has a typical organization for research articles.
Before the publication it requires significant improvements, especially:

  1. The section "Annotation" and "Conclusion" are almost identical, which is unacceptable. Please make corrections.
  2. It is necessary to significantly revise the section "Introduction", it is necessary to discuss the work on the creation of thermoelectric materials on a polymer basis, to note their advantages and disadvantages. It is also necessary to consider in more detail the use of CNTs and CNFs and their influence on various characteristics of polymer composites. I think the related references should be cited corresponding to each aspect, e.g. (but not limited to these): Polymers 202113(23), 4264; https://doi.org/10.3390/polym13234264: Polymers 2020, 12 (8), 1816, https://doi.org/10.3390/polym12081816; Polymers 202113(2), 278; https://doi.org/10.3390/polym13020278
  3. The article Polymer Journal 2021, 53, 443 1145-1152, doi: https://doi.org/10.1038/s41428-021-00518-7, that you published is very similar to the article that you submitted for publication in "Polymers". What is the difference between these articles, except that you are using different brands of CNFs? What is the scientific novelty of your new work? This should be detailed in the Introduction section.
  4. it is necessary to add more detailed information about the polymer matrix - give a table with the main physicochemical and technological properties in section 2.1.
  5. What is the reason for the choice of the amount of CNFs in the composite, why exactly 5 wt. %? In article [15], to which the authors refer, there is no such data and the content of more than 2.4 wt. % is not discussed. The choice of the number of CNFs must be justified.

Round 2

Reviewer 1 Report

Authors have addressed my major concerns about the manuscript in its initial form and I think that the revised version can be accepted for publication in Polymers.

Reviewer 2 Report

The authors did a good job to improve the article. I believe that article has become much better and now corresponds to the "Polymers" journal. I recommend this article for publication.